# Balanced Multi-Relational Graph Clustering

Zhixiang Shen
University of Electronic Science and
Technology of China
Chengdu, Sichuan, China
zhixiang.zxs@gmail.com

Haolan He
University of Electronic Science and
Technology of China
Chengdu, Sichuan, China
HaolanHe7777@gmail.com

Zhao Kang*
University of Electronic Science and
Technology of China
Chengdu, Sichuan, China
zkang@uestc.edu.cn

## Abstract

Multi-relational graph clustering has demonstrated remarkable success in uncovering underlying patterns in complex networks. Representative methods manage to align different views motivated by advances in contrastive learning. Our empirical study finds the pervasive presence of imbalance in real-world graphs, which is in principle contradictory to the motivation of alignment. In this paper, we first propose a novel metric, the Aggregation Class Distance, to empirically quantify structural disparities among different graphs. To address the challenge of view imbalance, we propose Balanced Multi-Relational Graph Clustering (BMGC), comprising unsupervised dominant view mining and dual signals guided representation learning. It dynamically mines the dominant view throughout the training process, synergistically improving clustering performance with representation learning. Theoretical analysis ensures the effectiveness of dominant view mining. Extensive experiments and in-depth analysis on real-world and synthetic datasets showcase that BMGC achieves state-of-the-art performance, underscoring its superiority in addressing the view imbalance inherent in multi-relational graphs. The source code and datasets are available at https://github.com/zxlearningdeep/BMGC.

## CCS Concepts

• **Computing methodologies** → **Cluster analysis**; *Neural networks*; *Regularization*.

## Keywords

Multi-view Graph Clustering, Imbalanced Graph Learning, Graph Representation Learning, Graph Homophily

**ACM Reference Format:**
Zhixiang Shen, Haolan He, and Zhao Kang. 2024. Balanced Multi-Relational Graph Clustering. In *Proceedings of the 32nd ACM International Conference on Multimedia (MM '24), October 28–November 1, 2024, Melbourne, VIC, Australia.* ACM, New York, NY, USA, 9 pages. https://doi.org/10.1145/3664647.3681325

---

*Corresponding author.

## 1 Introduction

Multi-relational graphs, which involve a set of nodes with multiple relations, are prevalent in the real world because of their extraordinary ability in characterizing complex systems [29]. Some typical instances are citation networks, social networks, and knowledge graphs [24, 38]. Recently, the unsupervised exploration of the inherent pattern in complex networks has attracted considerable attention, particularly in the context of multiview graph clustering (MVGC). Conventional MVGC techniques typically combine graph optimization with clustering techniques such as subspace clustering and spectral clustering [13, 22]. With the advancement of Graph Neural Networks (GNNs), a new wave of deep MVGC methods has been proposed, such as O2MAC [5], DMGI [25], HDMI [11], BTGF [28]. They have demonstrated significant efficacy.

However, representative MVGC methods often align all views to seek consistent information with the aid of a contrastive learning mechanism [11, 20, 28, 36]. This approach often ignores the fact that different views in real-world data do not always carry equal significance, i.e., the imbalance phenomenon. Our empirical analysis of real-world multi-relational graphs confirms this intuition. As shown in Fig. 1, different relations exhibit a big gap in classification accuracy. Therefore, naively aligning different views could degrade the final performance.

To this end, we address the view imbalance problem in multi-relational graphs in this work. Unlike other multiview data, the view differences in multi-relational graphs are rooted in their topology structures. Thus, a natural question arises: ($Q1$) **How can we quantify the structural disparities between views in multi-relational graphs?** Previous studies in multimodal learning indicate the presence of a dominant view in view-imbalanced data [34]. Given the clustering tasks, another question appears: ($Q2$) **How can we discover the dominant view without supervision to guide multi-relational graph clustering?**

In addressing $Q1$, we propose a simple yet effective view evaluation metric: Aggregation Class Distance (ACD). Unlike previous methods that solely calculate graph homophily ratios at the edge or node level [23], ACD takes into account both the aggregation process and the distribution of node classes. Empirical studies conducted on real-world datasets demonstrate the efficacy of this novel metric in evaluating the quality of views in multi-relational graphs.

For $Q2$, we propose Balanced Multi-Relational Graph Clustering (BMGC), which incorporates unsupervised dominant view mining and dual signal guided representation learning. A dynamic method of unsupervised exploration of the dominant view is employed throughout the training process, taking advantage of view-specific representations and original node features. Theoretical analysis establishes the connection between this approach and ACD, ensuring the effectiveness of dominant view mining. Afterward, dual signals

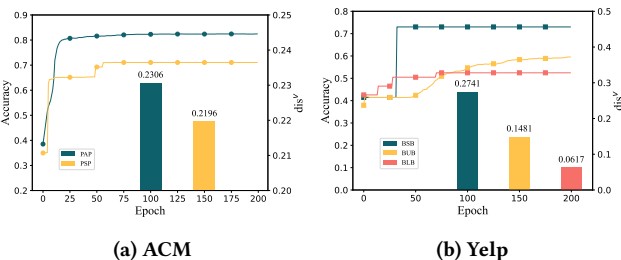

**(a) ACM**     **(b) Yelp**

**Figure 1: The node classification accuracy for each view of the test set, along with the corresponding ACD. The trends in accuracy and ACD are consistent.**

from the dominant view and node features supervise the graph embedding, promoting co-aligned representation learning. Finally, the dominant assignment is utilized to further enhance the clustering performance. In particular, dynamic extraction of the dominant view coupled with representation learning synergistically boosts model training.

We underscore the primary contributions of this study as follows:

- We explore view imbalance in multi-relational graphs and design a metric for evaluating view quality. Our empirical study confirms the presence of view disparities and validates the utility of our proposed metric.
- To our best knowledge, we are the first to tackle view imbalance in multi-relational graphs without supervision. BMGC integrates an unsupervised method for dominant view mining and dual signal guided representation learning. Furthermore, the dominant assignment is leveraged to facilitate self-training clustering. A theoretical guarantee is provided to demonstrate the efficacy of our approach.
- We conduct comprehensive experiments and in-depth analysis on real-world and synthetic datasets. BMGC surpasses existing advanced methods across all datasets, affirming its effectiveness and superiority in addressing view imbalance.

## 2 Related Work

### 2.1 Multiview Graph Clustering

Recently, there have been extensive explorations into multiview graph clustering. Typical shallow methods, such as MvAGC [13] and MCGC [22], combine graph filtering with self-expression learning to leverage attribute and structural information simultaneously.

With the progress in representation learning, several deep methods have emerged. Most of them start by unsupervised learning of node representations and then apply the k-means algorithm on these representations to obtain clustering results. O2MAC [5] is the first to employ GNN for MVGC, selecting the most informative view as input and reconstructing the graph structures of all views to capture shared information. Although O2MAC considers discrepancies in different graph structures, its encoding strategy, which retains only the best view, results in degradation into a single-view method. Moreover, it uniformly reconstructs the graph structures of all views, which in turn disregards the view imbalance, further leading to suboptimal results. DMGI [25] and HDMI [11] optimize

embeddings by maximizing mutual information between local and global representations. MGCCN [15], MGDCR [20], and BTGF [28] incorporate various contrastive losses to achieve the alignment of the representation and prevent dimension collapse. DuaLGR [14] extracts supervised signals from node attributes and graph structures to guide the MVGC. CoCoMG [26] and DMG [21] approach multi-view representation learning from the perspectives of consistency and complementarity. Although numerous methods achieve representation learning through multiview alignment, most of them overlook the inherent performance disparities between different views. These alignment-based methods tend to treat all graphs equally, which compromises the quality of node representations and thereby deteriorates the clustering results.

In supervised or semi-supervised tasks, methods like HAN [35] and SSAMN [30] consider the varying importance of views. However, they require labeled information for training, which is unsuitable for unsupervised tasks. **To our best knowledge, we are the first to address view imbalance in multiview graph clustering**.

### 2.2 Imbalanced Multiview Learning

Numerous efforts have been dedicated to addressing the challenges of imbalanced multiview learning from diverse perspectives. Works such as [8, 16] tackle imbalanced views through decision-level fusion. Specifically, they initially cluster each view and then fuse the view-specific clustering results. Another distinct avenue involves leveraging similarity graphs. MDcR [40] constructs balanced view-specific inter-instance similarity graphs, utilizes embedding techniques to acquire latent representations, and concatenates them to form the final representation for clustering. In contrast, FMUGE [39] takes a different approach to model order, initially combining view-specific similarity matrix to create a common similarity graph, followed by learning a comprehensive multiview representation. However, all of these methods cannot handle graph structure data.

Imbalanced multimodal learning has also attracted widespread attention. Recent theoretical advancements have demonstrated the potential of multimodal learning to surpass the upper limits of single-modal performance [10]. However, due to the varying confidence levels and noise across different modalities, the learning process is susceptible to inducing bias towards a dominant modality. To achieve a balanced multimodal classification, OGM [27] devises a modality-wise difference ratio to monitor the contribution discrepancy of each modality to the target, thus adaptively adjusting the gradients of each modality. Subsequently, PMR [6] proposes Prototypical Modality Rebalance, accelerating the slow-learning modality by enhancing its clustering towards prototypes. Despite the effectiveness of these methods, none have considered graph data. Therefore, methods to handle the view imbalance in the realm of unsupervised multiview graph learning are urgently needed.

## 3 Empirical Study

**Notation.** In this work, we define a multi-relational graph as $\mathcal{G} = \{\mathcal{V}, \mathcal{E}_1, \cdots, \mathcal{E}_v, \cdots, \mathcal{E}_V, X\}$, where $\mathcal{V}$ is the node set with $N$ nodes and $\mathcal{E}_v$ is the edge set in the $v$-th view. $V > 1$ is the number of relational graphs. $X \in \mathbb{R}^{N \times d_f}$ is the feature matrix and $x \in \mathbb{R}^N$ is a column of the feature matrix that represents a graph signal. $\tilde{A}^v$ denotes the original adjacency matrix of the $v$-th view.

$D^v$ represents the degree matrix. The normalized adjacency matrix of the $v$-th view is given by $A^v = (D^v)^{-\frac{1}{2}} \tilde{A}^v (D^v)^{-\frac{1}{2}}$. It is a well-known fact that the eigenvalues of $A^v$ in each view are contained within $[-1, 1]$. $\hat{A}^v = (D^v + I)^{-\frac{1}{2}} (\tilde{A}^v + I)(D^v + I)^{-\frac{1}{2}}$ represents the normalized adjacency matrix with a self-loop to each node, where $I$ is an identity matrix. $C$ is the number of node classes, and $y \in \mathbb{R}^N$ denotes the label vector.

In a multi-relational graph, the imbalance between views stems from differences in graph structure: some graphs contain more task-relevant information, while others are less task-relevant. Previous research has analyzed the impact of the graph structure on GNN from the perspective of graph homophily, suggesting that structures with high homophily ratios often exhibit superior performance [3, 42]. Here, the edge-level homophily ratio ($hr$) is defined as $hr = \frac{1}{|\mathcal{E}|} \sum_{(i,j) \in \mathcal{E}} \mathbb{1}(y_i = y_j)$, where $\mathbb{1}(\cdot)$ is the indicator function that equals 1 if its argument is true and 0 otherwise. However, recent studies have shown that neighbors of different classes may not necessarily make the nodes indistinguishable [19]. Graph structure analysis should consider node neighborhood patterns and the aggregation process. To quantify structural disparities across different views, we propose a simple yet effective metric: *Aggregation Class Distance* (ACD). ACD evaluates structure quality based on aggregated feature distribution of node classes, adapting better to real-world complexity than assuming a direct correlation between homophily ratio and task performance. The theoretical analysis in Section 5 substantiates this assertion. We choose the Simple Graph Convolution (SGC) as the aggregation method [37], a widely used representative aggregation operation [3, 19]. The ACD is defined as follows.

Definition 1. *The aggregation class distance for the $v$-th view, denoted as $dis^v$, is calculated as:*

$$X^v = (A^v)^K X, \quad \bar{X}_m^v = \frac{1}{N_m} \sum_{y_i = m} X_i^v \qquad (1)$$

$$dis^v = \frac{2}{C^2 - C} \sum_{m=1}^{C} \sum_{n=m+1}^{C} \|\bar{X}_m^v - \bar{X}_n^v\| \qquad (2)$$

*where $N_m$ is the number of nodes in class $m$ and $K$ denotes the radius of aggregation. The reciprocal of $\frac{2}{C^2-C}$ represents the computation count for pairwise inter-class distances.*

$\bar{X}_m^v$ represents the centroid of aggregated features for nodes with class $m$ in the $v$-th view. The metric $dis^v$ gauges the inter-class distance of aggregated features. A higher value indicates better discriminability among different classes.

To demonstrate the connection between ACD and view performance, we conduct empirical research on real-world datasets. We randomly select 30% of the nodes as the training set, leaving the remaining ones for the test set. The aggregation radius is set to 3, aligning with the common layer count of many GNN models [2]. A linear layer serves as the classifier. As shown in Fig. 1, the line represents the classification accuracy of each view, while the bar chart indicates the corresponding ACD. Different views yield different results, affirming the existence of view imbalance. In each dataset, the performance of one view significantly exceeds that of others, and we refer to it as the dominant view. Furthermore, views with higher ACD values exhibit better classification results,

underscoring the efficacy of ACD in gauging structural disparities between views. This empirical evidence supports the notion that ACD serves as a valuable metric for evaluating the quality of views in multi-relational graphs. It is worth noting that ACD uses node labels to assess the graph structure quality of each view, meaning that it is a supervised approach and cannot be directly applied to unsupervised learning tasks like clustering.

## 4 Methodology

In this section, we propose Balanced Multi-Relational Graph Clustering, as depicted in Fig. 2, to overcome inherent view imbalance.

### 4.1 Scalable Graph Encoding

Unlike most GNN-based approaches [20, 21], we decouple graph propagation and dimensionality reduction to improve scalability. Initially, we perform propagation on the node features separately for each view, acquiring view-specific aggregated features. Similarly to the approximate personalized propagation in [4], we introduce the features of the original node as a teleport vector in each layer of the propagation process:

$$X^{v,0} = X, \quad X^{v,k+1} = (1 - \alpha)\hat{A}^v X^{v,k} + \alpha X \qquad (3)$$

where $X$ acts as both the starting matrix and the teleport set for each view. The hyper-parameter $\alpha \in [0, 1]$ represents the teleport probability. $k \in [0, K - 1]$ and the aggregated features $X^v = X^{v,K}$. These features are then fed into a shared encoder for dimensionality reduction:

$$Z^v = f_\Theta(X^v) \qquad (4)$$

where $Z^v \in \mathbb{R}^{N \times d_r}$ denotes node representations in the $v$-th view. This decoupled setup avoids the time-consuming graph convolution operations during training. Subsequently, the representations for each view are fed into a shared decoder for the reconstruction of view-specific aggregated features. Effective training of each view is ensured by optimizing the reconstruction cosine error:

$$\hat{X}^v = g_\Theta(Z^v) \qquad (5)$$

$$\mathcal{L}_{REC} = \frac{1}{NV} \sum_{v=1}^{V} \sum_{i=1}^{N} \left(1 - \frac{(X_i^v)^\top \hat{X}_i^v}{\|X_i^v\| \cdot \|\hat{X}_i^v\|}\right) \qquad (6)$$

where the encoder $f_\Theta(\cdot)$ and the decoder $g_\Theta(\cdot)$ are both implemented using a Multilayer Perceptron (MLP) in this study.

### 4.2 Unsupervised Dominant View Mining

Due to the absence of label information, we cannot directly utilize ACD for the assessment of view quality in multi-relational graph clustering. Therefore, grounded in the principle of invariance in the distribution of similarity between instances, we propose an unsupervised method for dominant view mining:

$$v^* = \arg \min_v \|XX^\top - X^v(X^v)^\top\|_F^2 \qquad (7)$$

where $v^*$ denotes the dominant view. The above expression quantifies the discrepancy between the similarity matrices of original features and view-specific aggregated features, henceforth referred to as the "unsupervised metric". Therefore, the dominant view

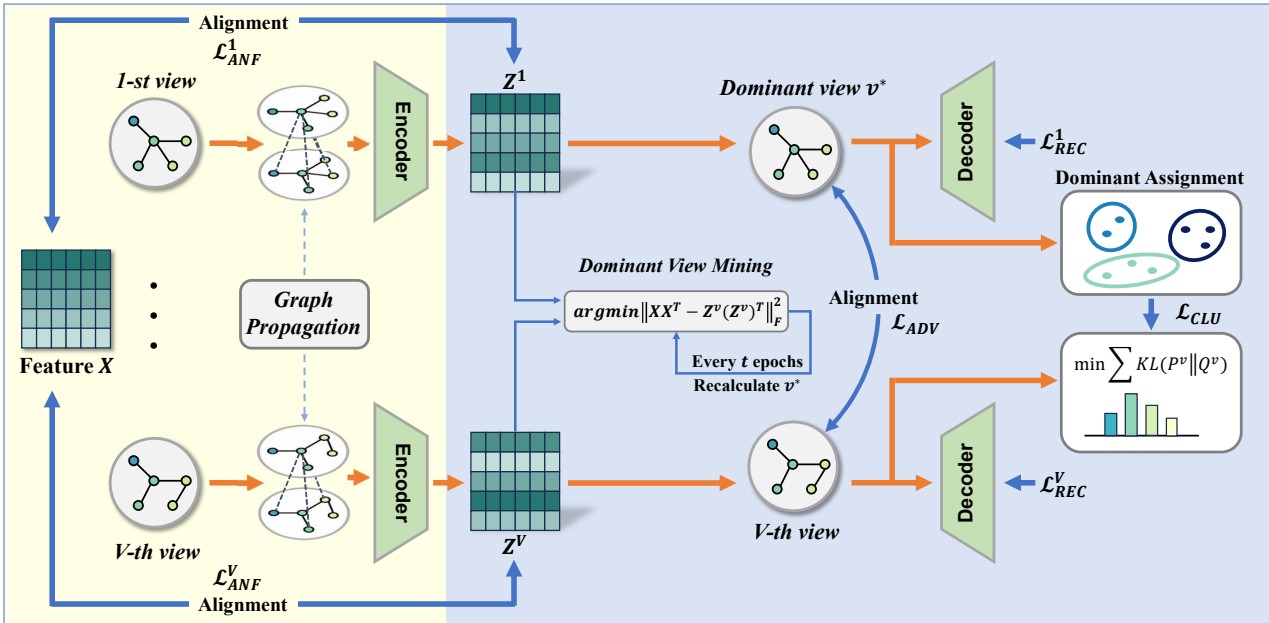

**Figure 2: Illustration of our proposed framework BMGC. Firstly, node representations for each view are obtained through scalable graph encoding. Then, unsupervised dominant view mining and dual signals guided representation learning synergistically facilitate model training. Finally, the dominant assignment further enhances clustering quality.**

should optimally maintain instance similarities. In Section 5, we theoretically establish the effectiveness of our approach.

Considering potential noise in real-world data, we refrain from directly using aggregated features to gauge view quality and, instead, rely on node representations:

$$v^* = \arg\min_v \|XX^\top - Z^v(Z^v)^\top\|_F^2 \tag{8}$$

In the training process, we initialize the dominant view using Equation (7) and recalculate it every $t$ epochs using Equation (8). As training progresses, the quality of node representations improves, thereby bolstering the reliability of dominant view mining. Simultaneously, the dominant view would guide representation learning, as elaborated later. It constitutes a mutually reinforcing process.

### 4.3 Co-aligned Representation Learning

After determining the dominant view, we use it to improve the representation quality. We employ contrastive learning to align the representations of other views with the dominant view. The representations of each view are projected to a shared latent space using separate learnable MLPs for fair similarity measurement and loss calculation. The contrastive loss is defined as follows:

$$\ell(Z_i^v, Z_i^{v^*}) = -\log \frac{e^{sim(\tilde{Z}_i^v, \tilde{Z}_i^{v^*})/\tau}}{\sum_{j=1}^N e^{sim(\tilde{Z}_i^v, \tilde{Z}_j^{v^*})/\tau}} \tag{9}$$

where $\tilde{Z}_i^v$ is the non-linear projection of $Z_i^v$. $sim(\cdot)$ refers to the cosine similarity and $\tau$ is the temperature parameter. The loss for aligning with the dominant view is given by:

$$\mathcal{L}_{ADV} = \frac{1}{2N(V-1)} \sum_{\substack{v=1 \\ v \neq v^*}}^V \sum_{i=1}^N (\ell(Z_i^v, Z_i^{v^*}) + \ell(Z_i^{v^*}, Z_i^v)) \tag{10}$$

Each view, along with the supervision from the dominant view, should also preserve a consistent similarity distribution among the nodes. Hence, we introduce a loss to ensure alignment with the node features:

$$\mathcal{L}_{ANF} = \frac{1}{N^2V} \sum_{v=1}^V \|XX^\top - Z^v(Z^v)^\top\|_F^2 \tag{11}$$

Note that the loss $\mathcal{L}_{ANF}$ shares a similar form with the unsupervised metric proposed in Section 4.2. This establishes a foundation for ensuring the reliability of our approach in continuously mining the dominant view during training. Ultimately, guided by dual signals from both the dominant view and node features, we accomplish the co-aligned representation learning:

$$\mathcal{L}_{CAL} = \mathcal{L}_{ADV} + \mathcal{L}_{ANF} \tag{12}$$

### 4.4 Dominant Assignment Enhanced Clustering

Most deep clustering methods leverage target distribution and soft cluster assignment probability distributions to achieve a self-training clustering scheme, with the cluster distribution typically obtained by applying k-means [17, 28, 31]. To improve the clustering performance, we substitute representation distributions in other views with cluster assignments derived from the dominant

view. Specifically, we apply k-means to the representations of the dominant view to obtain the dominant assignment:

$$\hat{y} = KMeans(\{Z_i^{v^*} : i = 1, \cdots, N\}) \tag{13}$$

Then, the soft assignment distribution $Q^v$ in the $v$-th view can be formulated as:

$$\sigma_j^v = \frac{1}{N_j} \sum_{\hat{y}_i = j} Z_i^v, \quad q_{ij}^v = \frac{(1 + \|Z_i^v - \sigma_j^v\|^2)^{-1}}{\sum_{k=1}^C (1 + \|Z_i^v - \sigma_k^v\|^2)^{-1}} \tag{14}$$

where $N_j$ denotes the number of nodes with the cluster assignment $j$ and $q_{ij}^v$ is measured using Student's t-distribution to denote the similarity between representation $Z_i^v$ and the clustering center $\sigma_i^v$. The target distribution $P^v$ is computed as:

$$p_{ij}^v = \frac{\left(q_{ij}^v\right)^2 / \sum_{i=1}^N q_{ij}^v}{\sum_{k=1}^C \left(\left(q_{ik}^v\right)^2 / \sum_{i=1}^N q_{ik}^v\right)} \tag{15}$$

We minimize the KL divergence between the distributions $Q^v$ and $P^v$ for each view to enhance cluster cohesion. The final node representation $Z = [Z^1, \cdots, Z^V] \in \mathbb{R}^{N \times V d_r}$ is obtained by concatenating representations from all views. We simultaneously minimize the KL divergence between the $Q$ and $P$ distributions of the final representation $Z$:

$$\mathcal{L}_{CLU} = KL(P\|Q) + \frac{1}{V} \sum_{v=1}^V KL(P^v\|Q^v) \tag{16}$$

The overall objective of BMGC, which we aim to minimize through the gradient descent algorithm, consists of three loss terms:

$$\mathcal{L} = \mathcal{L}_{REC} + \mathcal{L}_{CAL} + \mathcal{L}_{CLU} \tag{17}$$

For large-scale datasets, our method, which benefits from scalable graph encoding, eliminates the need for neighbor sampling during the training process. Consequently, we can directly perform mini-batch training, where all loss terms are computed solely from nodes within the batch.

## 5 Theoretical Analysis

In this section, we use a synthetic network to theoretically demonstrate the efficacy of BMGC in extracting the dominant view. For simplicity, we adopt SGC as the feature aggregation method.

***Data Assumption.*** A multi-relational graph $G$ has $N$ nodes partitioned into 2 equally sized communities $C_1$ and $C_2$. Let $c_1, c_2 \in \{0, 1\}^N$ be indicator vectors for membership in each community, that is, the $j^{th}$ entry of $c_i$ is 1 if the $j^{th}$ node is in $C_i$ and 0 otherwise. $G$ has $V$ views, each is generated by SBM [1], with intra- and inter-community edge probabilities $p^v$ and $q^v$. $G$ is such a graph model with a feature matrix $X = F + H \in \mathbb{R}^{N \times d_f}$, where each column of $H$ follows a zero-centered, isotropic Gaussian noise distribution $\mathcal{N}(0, \sigma^2 I)$ and these columns are mutually independent. The matrix $F$ is defined as $F = c_1 \mu_1^\top + c_2 \mu_2^\top$, where $\mu_1, \mu_2 \in \mathbb{R}^N$ has the same Euclidean norm $\|\mu\|$, representing the expected characteristic vector of each community. In addition, let $\bar{\mu} = \frac{1}{2}(\mu_1 + \mu_2)$ be the average of the feature vector means.

LEMMA 1. *Let $X^v$ be the aggregated feature matrix of the $v$-th view by applying SGC, with the number of hops $K$, to the expected adjacency matrix $\tilde{A}^v$ and the feature matrix $X$. Then, $X^v = F^v + c_1(\theta_1^v)^\top + c_2(\theta_2^v)^\top$, where $F^v = (\lambda_2^v)^K F + (1 - (\lambda_2^v)^K)(\mathbf{1}\bar{\mu}^\top)$, $\theta_1^v$ and $\theta_2^v \in \mathbb{R}^{d_f}$ are both distributed according to $\mathcal{N}(0, \frac{1}{N}(1 + (\lambda_2^v)^{2K})\sigma^2 I)$, and $\lambda_2^v = \frac{p^v - q^v}{p^v + q^v} \in [-1, 1]$ is the second largest non-zero eigenvalue of the associated normalized adjacency matrix $A^v$.*

THEOREM 1. *Let $\bar{X}_1^v$ and $\bar{X}_2^v$ denote the centroid of aggregated features for each community in the $v$-th view. Then, $\mathbb{E}\left[\bar{X}_1^v - \bar{X}_2^v\right] = (\lambda_2^v)^K(\mu_1 - \mu_2)$ and $\mathbb{E}\left[XX^\top - X^v(X^v)^\top\right] = \frac{1 - (\lambda_2^v)^{2K}}{2}(\|\mu\|^2 - \mu_1^\top \mu_2)(c_1 c_1^\top + c_2 c_2^\top - c_1 c_2^\top - c_2 c_1^\top) + \omega(\sigma^2)$, where $\omega(\sigma^2)$ represents the sum of terms containing $\sigma^2$ that are of negligible magnitude.*

When negligible terms are ignored, it becomes clear that the unsupervised identification of the dominant view essentially corresponds to the view with the maximum $(\lambda_2^v)^2$. Additionally, under this data assumption, $dis^v = \|\bar{X}_1^v - \bar{X}_2^v\|$ indicates that a view with a larger $(\lambda_2^v)^2$ is more likely to exhibit a larger ACD. Specifically, the consistent changes in both $dis^v$ and $\|XX^\top - X^v(X^v)^\top\|_F^2$ related to $(\lambda_2^v)^2$ reveal that the identified dominant view is the one with the largest ACD value.

The theoretical findings are interpretable. $\lambda_2^v$ is influenced by the ratio of intra- to inter-community edge probabilities, essentially measuring the homophily level of the graph structure in the $v$-th view. It tends toward 1 for complete homophily and $-1$ for complete heterophily. In both cases where $(\lambda_2^v)^2$ approaches 1, there would be no confusion between nodes from different communities. For example, when $\lambda_2^v = -1$, if $K$ is odd, feature exchange occurs between the two communities after aggregation; if $K$ is even, the features would remain unchanged. In such cases with pure graph structures, the view consistently demonstrates a larger ACD and a smaller unsupervised metric, while the homophily ratio would tend toward zero as $\lambda_2^v$ approaches $-1$. On the contrary, when $\lambda_2^v$ nears 0, the graph structure becomes uninformative, resulting in the view with a smaller ACD and a larger unsupervised metric. In summary, we draw two conclusions: **1. ACD is more universally applicable than the traditional homophily ratio in assessing the relevance between graph structures and downstream tasks. 2. The dominant view, mined through our unsupervised method, essentially corresponds to the view with the maximum ACD value, which ensures the effectiveness of unsupervised dominant view mining.**

## 6 Experiments

### 6.1 Datasets and Metrics

***Datasets.*** We employ five publicly available real-world benchmark datasets and a large-scale dataset. ACM [5], ACM2 [7], and DBLP [41] are citation networks. Yelp [18] and Amazon [9] are review networks. MAG [33] is a large-scale citation network, constituting the largest dataset in multi-relation graph clustering thus far.

***Metrics.*** We adopt four popular clustering metrics, including Accuracy (ACC), Normalized Mutual Information (NMI), F1 score, and Adjusted Rand Index (ARI). A higher value of them indicates a better performance.

**Table 1: Clustering results on real-world datasets. The best and second-place results are highlighted using bold and underline, respectively. The asterisk (*) denotes the supervised baseline.**

| Datasets | Metric | HAN* 2019 | VGAE 2016 | DGI 2019 | O2MAC 2020 | DMGI 2020 | MvAGC 2021 | HDMI 2021 | MCGC 2021 | MGDCR 2023 | DuaLGR 2023 | DMG 2023 | BTGF 2024 | BMGC |
|---|---|---|---|---|---|---|---|---|---|---|---|---|---|---|
| Amazon | NMI | 0.4037 | 0.0163 | 0.0532 | 0.1344 | 0.2623 | 0.2322 | 0.3702 | 0.2149 | 0.0318 | 0.2767 | 0.1218 | 0.3853 | **0.5768** |
| | ARI | 0.4241 | 0.0129 | 0.0202 | 0.0898 | 0.2605 | 0.1141 | 0.2735 | 0.1056 | 0.0055 | 0.2715 | 0.0283 | 0.2829 | **0.5626** |
| | ACC | 0.7437 | 0.3194 | 0.3762 | 0.4428 | 0.5581 | 0.5188 | 0.5251 | 0.4683 | 0.3489 | 0.6123 | 0.3887 | 0.6603 | **0.7856** |
| | F1 | 0.7433 | 0.2725 | 0.2859 | 0.4424 | 0.5463 | 0.5072 | 0.5448 | 0.4804 | 0.2039 | 0.6215 | 0.3441 | 0.6612 | **0.7851** |
| ACM | NMI | 0.6864 | 0.491 | 0.6364 | 0.6923 | 0.6441 | 0.6735 | 0.645 | 0.7126 | 0.721 | 0.7328 | 0.7561 | 0.758 | **0.7841** |
| | ARI | 0.7489 | 0.5448 | 0.6822 | 0.7394 | 0.6729 | 0.7212 | 0.674 | 0.7627 | 0.6496 | 0.7942 | 0.8033 | 0.8085 | **0.8329** |
| | ACC | 0.9088 | 0.8228 | 0.8816 | 0.9042 | 0.8724 | 0.8975 | 0.874 | 0.9147 | 0.919 | 0.9271 | 0.9302 | 0.9322 | **0.9413** |
| | F1 | 0.9085 | 0.8231 | 0.8829 | 0.9053 | 0.8709 | 0.8986 | 0.872 | 0.9155 | 0.8678 | 0.927 | 0.9306 | 0.9331 | **0.9416** |
| DBLP | NMI | 0.6998 | 0.6934 | 0.6168 | 0.7294 | 0.7489 | 0.7723 | 0.6361 | 0.6561 | 0.7595 | 0.7559 | 0.7907 | 0.6027 | **0.8013** |
| | ARI | 0.7641 | 0.7413 | 0.5653 | 0.7783 | 0.8032 | 0.828 | 0.6145 | 0.7088 | 0.8072 | 0.8168 | 0.8384 | 0.6534 | **0.8539** |
| | ACC | 0.9015 | 0.8868 | 0.7446 | 0.9071 | 0.9159 | 0.9284 | 0.7832 | 0.8752 | 0.9182 | 0.9242 | 0.9344 | 0.8509 | **0.9401** |
| | F1 | 0.8966 | 0.8748 | 0.7392 | 0.901 | 0.9075 | 0.9231 | 0.7372 | 0.8186 | 0.9123 | 0.918 | 0.9303 | 0.8456 | **0.9364** |
| ACM2 | NMI | 0.6435 | 0.4507 | 0.5779 | 0.4223 | 0.574 | 0.1819 | 0.5902 | 0.5307 | 0.5447 | 0.5988 | 0.6341 | 0.6483 | **0.7285** |
| | ARI | 0.6979 | 0.4347 | 0.5174 | 0.4451 | 0.5243 | 0.1879 | 0.5472 | 0.4396 | 0.4372 | 0.6399 | 0.6726 | 0.6776 | **0.7601** |
| | ACC | 0.8943 | 0.7358 | 0.8114 | 0.7537 | 0.8148 | 0.5949 | 0.8258 | 0.7129 | 0.6838 | 0.8676 | 0.8796 | 0.8853 | **0.9185** |
| | F1 | 0.8955 | 0.7101 | 0.8261 | 0.7418 | 0.8267 | 0.4484 | 0.8386 | 0.5809 | 0.5854 | 0.8653 | 0.8773 | 0.8887 | **0.9215** |
| Yelp | NMI | 0.6762 | 0.3919 | 0.3942 | 0.3902 | 0.3729 | 0.2439 | 0.3912 | 0.3835 | 0.4423 | 0.6621 | 0.391 | 0.4135 | **0.7173** |
| | ARI | 0.7205 | 0.4257 | 0.4262 | 0.4253 | 0.3418 | 0.2925 | 0.3922 | 0.3517 | 0.4647 | 0.6847 | 0.4261 | 0.3564 | **0.7381** |
| | ACC | 0.9082 | 0.6507 | 0.6529 | 0.6507 | 0.5893 | 0.6314 | 0.6452 | 0.6561 | 0.7271 | 0.8948 | 0.6512 | 0.7192 | **0.9151** |
| | F1 | 0.9163 | 0.5674 | 0.5679 | 0.5674 | 0.4878 | 0.567 | 0.5874 | 0.5749 | 0.5443 | 0.9051 | 0.5679 | 0.7307 | **0.9246** |

## 6.2 Experimental Setup

**Baselines.** We compare BMGC with various baselines, including the supervised multiview graph method HAN [35], single-view graph clustering methods VGAE [12] and DGI [32], and multiview graph clustering methods O2MAC [5], DMGI [25], MvAGC [13], HDMI [11], MCGC [22], MGDCR [20], DuaLGR [14], DMG [21], and BTGF [28]. All other methods are unsupervised excluding HAN, which serves as the supervised baseline.

**Parameter Setting.** Our model is trained for 400 epochs using the Adam optimizer with a learning rate of 1e-2. The weight decay of the optimizer is set to 1e-4. The recalculation interval $t$ for the dominant view is every 50 epochs. We set the representation dimension $d_r$ to 64 for ACM2 dataset and 10 for the other datasets. The temperature parameter $\tau$ is fixed at 1. The radius of graph propagation, $K$, is fixed at 3 and the teleport probability $\alpha$ is tuned in $[0, 0.3, 0.5]$. All experiments are implemented on the PyTorch platform using an Intel(R) Xeon(R) Platinum 8352V CPU and a GeForce RTX 4090 24G GPU.

## 6.3 Evaluation on Real-world Datasets

To evaluate the performance of our model, we compare BMGC with multiple baselines on five real-world datasets. For the supervised baseline HAN, we employ k-means on the node embeddings of the test set to yield clustering results. We conduct single-view clustering methods separately for each view and present the best results. Generally, BMGC consistently outperforms all compared methods regarding four metrics over all datasets. From Table 1, we have the following observations:

- The advantages of BMGC become evident when compared to other methods. In particular, our model significantly outperforms existing methods, including the supervised baseline, on Amazon and ACM2 datasets. Regarding second-place results on Amazon, our model improves NMI and ARI by 42.9% and 32.7%, respectively.

- In general, multiview graph methods outperform single-view methods like VGAE and DGI, demonstrating the superiority of multiview methods in graph clustering. However, in the Yelp dataset, most multiview baselines underperform compared to single-view methods, which may be attributed to the fact that these multiview methods overlook the imbalance among different views, leading to worse performance. Moreover, while the supervised baseline HAN surpasses the unsupervised baselines on most datasets, BMGC still outperforms it, underscoring the superiority of our method.

- Our model outperforms O2MAC which considers information differences among views. O2MAC retains only the most informative view while discarding others, which to some extent degenerates into a single-view method with worse results. Our model uses all the views and achieves better results by aligning the other views with the dominant view.

## 6.4 Evaluation on Synthetic Datasets

To further compare BMGC with other methods in addressing the imbalanced problem, we introduce a new synthetic dataset based on cSBM [4], named multi-relational cSBM. The multi-relational cSBM initially generates three views, each possessing unique graph structures with uniform homophily ratios, and sharing a common feature

Table 2: Clustering results on synthetic datasets.

| Perturbation ratios | Metric | SGC | | | DMGI 2020 | HDMI 2021 | MGDCR 2023 | DuaLGR 2023 | DMG 2023 | BTGF 2024 | BMGC |
|---|---|---|---|---|---|---|---|---|---|---|---|
| | | view 1 | view 2 | view 3 | | | | | | | |
| 20% | NMI | 0.6142 | 0.5191 | 0.4973 | 0.675 | 0.5869 | 0.8702 | 0.4065 | 0.6326 | 0.7874 | **0.9209** |
| | ARI | 0.7207 | 0.6278 | 0.6053 | 0.7765 | 0.694 | 0.9293 | 0.5074 | 0.7321 | 0.8697 | **0.9612** |
| | ACC | 0.9245 | 0.8962 | 0.8891 | 0.9406 | 0.9165 | 0.982 | 0.8562 | 0.9278 | 0.9663 | **0.9902** |
| 50% | NMI | 0.6142 | 0.3956 | 0.3896 | 0.6425 | 0.4766 | 0.7959 | 0.3314 | 0.5748 | 0.7213 | **0.8913** |
| | ARI | 0.7207 | 0.4959 | 0.4888 | 0.7471 | 0.583 | 0.8761 | 0.4229 | 0.683 | 0.8162 | **0.9432** |
| | ACC | 0.9245 | 0.8521 | 0.8496 | 0.9322 | 0.8818 | 0.968 | 0.8252 | 0.9132 | 0.9517 | **0.9856** |
| 100% | NMI | 0.6142 | 0.2514 | 0.267 | 0.573 | 0.2821 | 0.6887 | 0.322 | 0.5195 | 0.6505 | **0.8178** |
| | ARI | 0.7207 | 0.327 | 0.3457 | 0.6812 | 0.3646 | 0.7885 | 0.4115 | 0.6257 | 0.7545 | **0.8926** |
| | ACC | 0.9245 | 0.7859 | 0.7941 | 0.9127 | 0.8019 | 0.944 | 0.8208 | 0.8955 | 0.9343 | **0.9724** |
| 150% | NMI | 0.6142 | 0.1679 | 0.1564 | 0.4372 | 0.0374 | 0.5711 | 0.3079 | 0.4659 | 0.6104 | **0.7821** |
| | ARI | 0.7207 | 0.223 | 0.2086 | 0.5362 | 0.0441 | 0.6789 | 0.3953 | 0.5657 | 0.7175 | **0.8656** |
| | ACC | 0.9245 | 0.7361 | 0.7284 | 0.8662 | 0.5948 | 0.9121 | 0.8144 | 0.8761 | 0.9235 | **0.9652** |

matrix. All nodes are categorized into two classes. We randomly add noisy edges to two of these graphs to induce perturbations, where the perturbation ratio $\rho$ controls the proportion of randomly added edges, simulating the imbalanced multi-relational graph. The undisturbed view is denoted as view 1, representing the dominant view, while the other two views are referred to as view 2 and view 3. Experiments are carried out for four $\rho$ values: [20%, 50%, 100%, 150%].

To reveal the performance discrepancies of different views, we use SGC on each view to obtain view-specific aggregated features and then obtain clustering results through k-means. We select several representation learning-based approaches for comparison. The results, as shown in Table 2, indicate that a higher perturbation ratio leads to poorer performance for all methods. Our detailed observations are as follows.

First, the majority of multiview graph clustering methods yield unsatisfactory results. As the perturbation ratio increases, their performance degrades to a lower level. Second, as the perturbation ratio reaches 150%, the performance of the comparative methods even drops below the SGC result in view 1, indicating that when there is extensive noise in certain views of the dataset, the performance of multiview methods may deteriorate compared to the single view methods. In our method, aligning with the dominant view prevents the result from being impaired by low-quality views with noise. Third, our model maintains relatively stable performance as the perturbation ratio increases, with a maximal variation range of 17.7%, 11%, and 2.6% for NMI, ARI, and ACC respectively, showcasing the robustness for noisy data.

## 6.5 Evaluation on Large-scale Dataset

To evaluate the efficiency of BMGC, we conduct experiments on a large-scale multi-relational graph MAG. We select some scalable representation learning-based methods as baselines, while the remaining models run out of memory. We set the representation

Table 3: Quantitative results with standard deviation (% ± $\sigma$) and execution time (seconds) on MAG.

| Methods | NMI | ARI | ACC | F1 | Time |
|---|---|---|---|---|---|
| k-means | 42.04 | 32.34 | 58.63 | 59.81 | - |
| DGI | 53.56±0.48 | 42.60±0.83 | 59.89±1.10 | 57.17±1.88 | 36 |
| DMGI | 49.71±1.37 | 38.91±1.35 | 53.57±0.54 | 49.59±1.39 | 118 |
| HDMI | 48.15±0.98 | 34.92±1.27 | 51.78±1.37 | 49.80±2.04 | 105 |
| MGDCR | 54.43±1.17 | 43.98±1.16 | 61.37±2.46 | 60.53±3.19 | 39 |
| DMG | 44.04±3.32 | 36.97±2.86 | 57.65±2.03 | 55.32±2.53 | 95 |
| **BMGC** | **57.01±0.19** | **47.84±0.27** | **65.31±1.25** | **63.68±1.84** | **25** |

dimension to 128 and the batch size to 5000. Table 3 presents the results with standard deviation and training time. Due to our scalable graph encoding that eliminates time-consuming neighbor sampling and graph convolution operations during training, BMGC achieves optimal results with the shortest training time.

In summary, across all datasets, BMGC consistently exhibits superior performance. The stable results obtained in these experiments demonstrate the effectiveness of our methods in addressing the view imbalance of multi-relational graphs.

## 6.6 Ablation Study

To validate the effectiveness of different components in our model, we compare the performance of BMGC with its three variants:

- Employing BMGC without $\mathcal{L}_{ADV}$ to show the significance of alignment with the dominant view.
- Employing BMGC without $\mathcal{L}_{ANF}$ to observe the impact of alignment with the node features.
- Employing BMGC without $\mathcal{L}_{CLU}$ to reveal the influence of the dominant assignment on clustering performance.

Based on Table 4, we can draw the following conclusions. First, the results of BMGC are better than all variants, indicating that

**Table 4: Performance of BMGC and its variants.**

| Variants | Amazon | | ACM | | DBLP | | ACM2 | | Yelp | |
|---|---|---|---|---|---|---|---|---|---|---|
| | NMI | ACC | NMI | ACC | NMI | ACC | NMI | ACC | NMI | ACC |
| **BMGC** | **0.5768** | **0.7856** | **0.7841** | **0.9413** | **0.8013** | **0.9401** | **0.7285** | **0.9185** | **0.7173** | **0.9151** |
| w/o $\mathcal{L}_{ADV}$ | 0.5303 | 0.7534 | 0.7366 | 0.9261 | 0.7773 | 0.9314 | 0.6054 | 0.8276 | 0.6763 | 0.8955 |
| w/o $\mathcal{L}_{ANF}$ | 0.4234 | 0.6452 | 0.7667 | 0.9368 | 0.7923 | 0.9349 | 0.6762 | 0.8816 | 0.7041 | 0.9139 |
| w/o $\mathcal{L}_{CLU}$ | 0.5629 | 0.7794 | 0.7726 | 0.9371 | 0.7857 | 0.9346 | 0.7263 | 0.7477 | 0.6864 | 0.9052 |

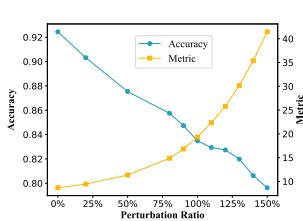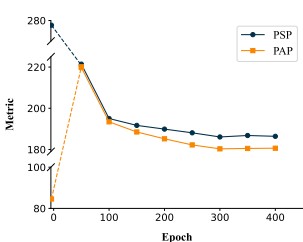

(a) The effectiveness of unsupervised dominant view mining.

(b) The reliability of dynamic mining in the training process.

**Figure 3: Case study on synthetic (left) and ACM (right) datasets. The specific meanings of the "Metric" in each figure can be found in Section 6.7.**

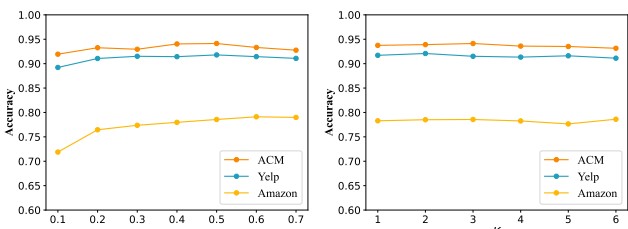

**Figure 4: The influence of $\alpha$ (left) and $K$ (right).**

all components are critical to our model. Second, the loss of alignment with the dominant view ($\mathcal{L}_{ADV}$) seems to make the most contribution to the results, while the loss for alignment with the node features ($\mathcal{L}_{ANF}$) contributes more to Amazon. This could be attributed to the universally subpar quality of all graph structures within the Amazon dataset, thereby amplifying the importance of node features. Additionally, the dominant assignment ($\mathcal{L}_{CLU}$) indeed improves clustering performance.

## 6.7 Case Study

**Effectiveness of Unsupervised Mining.** We delve deep into examining the impact of the perturbation ratio on both the accuracy and the unsupervised metric ($\|XX^{\top} - X^v(X^v)^{\top}\|_F^2 / N$) in View 3 of the synthetic dataset. As depicted in Fig. 3a, it is conspicuous that with the increase of the perturbation ratio from 0 to 150%, the accuracy consistently decreases, indicating a continuous decline in view quality. In parallel, the corresponding unsupervised metric indeed rises. This highlights the effectiveness of our unsupervised dominant view extraction method, aligning with the conclusions drawn in the theoretical analysis in Section 5.

**Reliability of Dynamic Mining.** To provide a detailed description of the process through which our model uncovers the dominant view, we demonstrate the evolution of the unsupervised metric ($\|XX^{\top} - Z^v(Z^v)^{\top}\|_F^2 / N$), used to mine the dominant view, for each view of the ACM dataset. As illustrated in Fig. 3b, the data points on the y-axis represent the aggregated node features used to initialize the dominant view. These points are connected by dashed lines to subsequent data points derived from node representations. Throughout the training, the metrics of both views decrease, and

the PAP view consistently exhibits lower values compared to the PSP view. This consistent trend indicates that PAP emerges as the dominant view, aligning seamlessly with our empirical analysis. After 250 epochs, the metrics converge. This result emphasizes the reliability of dynamically excavating the dominant view.

## 6.8 Hyper-parameters Study

We conduct a hyper-parameter analysis on the teleport probability $\alpha$ and the radius of graph propagation $K$ on three datasets ACM, Yelp, and Amazon. The result is given in Fig. 4. From the figure on the left, we can observe that our model shows low sensitivity to the change of $\alpha$. However, when $\alpha$ is too low, the performance shows a noticeable decrease. This can be attributed to the fact that the lower value of $\alpha$ leads to a decreased influence of the features of the original nodes in the propagation process. In the figure on the right, we can see that the performance is stable to the change of $K$. Notable performance can be achieved when $K$ is small, improving the efficiency of the model in practical applications.

## 7 Conclusion

In this study, we thoroughly investigate the prevalent challenge of view imbalance in real-world multi-relational graphs. We introduce a novel metric, the Aggregation Class Distance, to empirically quantify structural disparities among different graphs. To tackle view imbalance, we propose Balanced Multi-Relational Graph Clustering, which dynamically mines the dominant view throughout the training process, collaborating with representation learning to enhance clustering performance. Theoretical analysis validates the efficacy of unsupervised dominant view mining. Extensive experiments and in-depth analysis on both real-world and synthetic datasets consistently demonstrate the superiority of our model over existing state-of-the-art methods.

## Acknowledgments

This work was supported by the National Natural Science Foundation of China (No. 62276053).

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
