# OpenReview forum: "Balanced Multi-Relational Graph Clustering"
_acmmm.org/ACMMM/2024/Conference — MM2024 Poster_

### Official Review · Reviewer_TZzr · 2024-05-24

**Rating:** 1
**Confidence:** 3

**Summary:**

In this paper, authors address the issue of imbalanced data in multi-relational graphs by proposing a Balanced Multi-Relational Graph Clustering (BMGC) method. Additionally, authors introduce a novel metric to measure structural differences across various graphs. Theoretical analysis demonstrates the effectiveness of the proposed dominant view mining. Extensive experiments validate that BMGC achieves optimal performance.

**Strengths:**

1, This paper attempts to provide a new metric for measuring graph balance.
2, The method's performance was evaluated on multiple datasets during the experiments.

**Limitations:**

1, The problem of imbalance in multi-relational graphs is not clearly defined. Accuracy is influenced by multiple factors, and merely pointing to low accuracy as evidence of data imbalance is insufficient to substantiate the issue.

2, The authors use performance to evaluate the effectiveness of ACD in measuring structural differences between views, which is inadequate to prove that ACD is a valid metric for assessing graph balance. Additionally, the authors claim in lines 329-332 that ACD is a supervised method, while in lines 562-564, they use it alongside unsupervised metrics to evaluate the effectiveness of the unsupervised dominant graph. It is illogical to use a supervised metric to assess the effectiveness of an unsupervised graph.

3, In section 4.1, the authors claim their method enhances scalability. What is the relevance of this claim to the problem addressed in the paper? Furthermore, there is no experimental evidence provided to substantiate the claim of improved scalability.

4, Some references need updating, such as [27], which has been published. Additionally, the reference format is inconsistent, as seen with [2], [3], and [4].

5, The readability of the figures is poor. For instance, in Figure 1(a), the x-axis tick values are incomplete, and the epoch is not labeled. Both Figure 1(a) and Figure 1(b) lack labels for the ACD metric.

6, The readability of the paper is suboptimal. It is recommended to revision the manuscript.

**Suitability:**

2

---

### Official Review · Reviewer_4T42 · 2024-05-24

**Rating:** 3
**Confidence:** 4

**Summary:**

The paper addresses the issue of view imbalance in multi-relational graphs, where nodes are interconnected by multiple types of relationships. The authors introduce the Aggregation Class Distance (ACD) metric to quantify structural disparities among views. They propose the Balanced Multi-Relational Graph Clustering (BMGC) method, which dynamically identifies the dominant view during training and uses it to guide representation learning. This approach includes unsupervised dominant view mining and dual signal guided learning to enhance clustering performance. BMGC's scalable graph encoding improves computational efficiency. Extensive experiments on real-world and synthetic datasets demonstrate that BMGC consistently outperforms existing methods, confirming its effectiveness.

**Strengths:**

1.	The paper introduces the Aggregation Class Distance (ACD) metric to quantify structural disparities among different views in multi-relational graphs
2.	The paper provides a theoretical foundation for the proposed methods, offering guarantees for the effectiveness of the dominant view mining.
3.	The extensive empirical evaluation on both real-world and synthetic datasets demonstrates BMGC's consistent outperformance over existing state-of-the-art methods

**Limitations:**

4.	This paper leaks core novel contributions, as the proposed method appears to be a combination of existing techniques or easily obtainable methods..
5.	The loss function, which incorporates full-size negative samples in the contrastive learning objective, requires three iterations of similarity computation. This makes the approach time-consuming, especially given that the proposed method operates on the entire graph rather than employing batch-wise optimization.
6.	The experimental analysis section lacks essential details. For instance, in Section 6.5, the execution time experiments do not report the experimental environment for the baseline comparisons. Additionally, the standard deviations of the experimental results are not provided, which is crucial for assessing the robustness.

**Suitability:**

2

---

### Official Review · Reviewer_GFkm · 2024-05-25

**Rating:** 6
**Confidence:** 4

**Summary:**

This paper finds the imbalance issue of multi-relational graphs and defines a new metric to measure it. To solve this problem, the authors propose a novel clustering method BMGC, which consists of unsupervised dominant view mining and dual signals guided representation learning. Theoretical and experimental analysis verify its effectiveness. Overall, the technical contributions are solid and sound.

**Strengths:**

1.The found imbalance phenomenon is quite interesting, which inherently limits the performance of existing graph clustering methods. I believe this finding opens a new direction for improving graph clustering performance in the future.
2.The proposed BMGC method is novel. Theoretical analysis shows its effectiveness. The approach is significant in the Multiview clustering community.
3.Comprehensive experiments on synthetic and real graphs show the effectiveness and efficiency of the proposed method, which are convincing to me.

**Limitations:**

1.The Algorithm is missing. It is unclear how the objective function (17) is solved. Adding and describing an algorithm for the entire workflow would strengthen this paper.
2.The writing could be improved and carefully checked. For example, “Most of them start by unsupervised learning of node representations and -> Most of them start with unsupervised learning of node representations and”,” Recent theoretical advancements have demonstrated the potential-> Recent theoretical advances have demonstrated the potential”.
3.Some references are missing. For example, the reconstruction cosine loss in Eq.(6) should be cited. It is derived from: Graphmae: Self-supervised masked graph autoencoder.

I also have some questions.
1. What is the difference between the two terms in Eq.10? More explanations are desired.
2. What does the 'metric' represent on the y-axis in Figure 3?
3. Let's suppose that a multi-relational graph has three or more views; could it happen that two views have the same ACD score? Why? Could you please give more details regarding this case?

**Suitability:**

3

---

### Official Review · Reviewer_cJt1 · 2024-05-25

**Rating:** 6
**Confidence:** 4

**Summary:**

To address the question regarding the importance of the views in a multi-relational graph, the authors proposed a new metric, Aggregation Class Distance (ACD), which is utilized to determine the most dominant view. ACD is used to empirically quantify structural disparities among different views of the multi-relational graph while considering the node features (attributes) and their corresponding structural connectivity (edges in adjacency matrix). Based on it, a novel clustering method and theoretical analysis are provided.

**Strengths:**

1. The idea is interesting and novel. The proposed method is theoretically guaranteed.
2. The paper is well-organized and articulates the problem statement, methodology, and findings clearly. The introduction of new concepts like ACD and the explanation of the BMGC framework are coherent. In particular, the figures improve the paper presentation.
3.The authors have conducted experiments to show different perspectives on their proposal and provided a link to the source code.

**Limitations:**

1. The objective function is a combination of three terms. However, there is no hyperparameters, which one is more important?
2. More descriptions about the design of ACD and the results in other evaluated datasets could be given.
3. It would be better is the clustering results from different methods are visualized.

**Suitability:**

3

---

### Official Review · Reviewer_yDzV · 2024-05-27

**Rating:** 3
**Confidence:** 3

**Summary:**

The paper addresses the issue of view imbalance in multi-relational graphs and proposes a new method to enhance clustering performance. The BMGC framework introduces the Aggregation Class Distance (ACD) metric to quantify structural disparities among different graph views. It employs unsupervised dominant view mining and dual signals guided representation learning to dynamically identify the dominant view and improve clustering results. Extensive experiments on real-world and synthetic datasets demonstrate the effectiveness of BMGC, claiming it outperforms existing methods.

**Strengths:**

- The unsupervised method for identifying the dominant view dynamically throughout training is innovative and addresses a crucial challenge in multi-relational graph clustering.
- The paper provides extensive experimental results on both real-world and synthetic datasets, showcasing the robustness and superiority of BMGC over existing methods.

**Limitations:**

- How robust is the unsupervised dominant view mining method to variations in the quality of different views?
- Can the authors provide more detailed empirical validation of the ACD metric in various graph settings? How does ACD compare with existing metrics in terms of effectiveness and reliability?
- What are the impacts of different hyperparameter settings on the performance of BMGC? Are there specific guidelines for selecting these parameters?
- Can the authors provide more detailed results or case studies on the scalability of BMGC, especially with significantly larger datasets and more complex graph structures?
- Please provide the variance of the main results to demonstrate the robustness of the method.
- Please provide the code link to enhance the persuasiveness of the results.

**Suitability:**

2

---

### Meta-Review · Area_Chair_P8Yz · 2024-06-29

**Recommendation:** Accept (Poster)
**Confidence:** 4

**Metareview:**

This paper received a mixed rating from reviewers. From their reviewers, although this paper has some merits, most of the reviewers are not convinced with the major concerns including experimental analysis and novelty. I have also checked the concerns raised by the reviewers carefully, especially Reviewers 4T42 and yDzV. Overall, the paper stands below the borderline. I'm inclined to reject the paper and encourage the authors to revise the paper for future venues.

***TPC Addendum***
The paper has received a split verdict and is at the borderline of acceptance. Given two very strong positive votes and some improvement in scores after rebuttal, the TPC are suggesting an acceptance to allow for the debate and the conversation to continue at the conference.